# Dietary Fibre Modulates Gut Microbiota in Late Pregnancy Without Altering SCFA Levels, and Propionate Treatement Has No Effect on Placental Explant Function

**DOI:** 10.3390/nu17071234

**Published:** 2025-04-01

**Authors:** Chelsea L. Vanderpeet, Emily S. Dorey, Elliott S. Neal, Thomas Mullins, David H. McIntyre, Leonie K. Callaway, Helen L. Barrett, Marloes Dekker Nitert, James S. M. Cuffe

**Affiliations:** 1School of Biomedical Sciences, The University of Queensland, Brisbane, QLD 4072, Australia; c.vanderpeet@uq.edu.au (C.L.V.); e.neal@uq.edu.au (E.S.N.); thomas.mullins@uq.net.au (T.M.); 2Mater Research Institute, The University of Queensland, South Brisbane, QLD 4101, Australia; emily.dorey@mater.uq.edu.au (E.S.D.); h.d.mcintyre@uq.edu.au (D.H.M.); helen.barrett1@health.nsw.gov.au (H.L.B.); 3Mater Hospital Brisbane, South Brisbane, QLD 4101, Australia; 4Department of Obstetric Medicine, Royal Brisbane and Women’s Hospital, Herston, QLD 4059, Australia; leonie.callaway@health.qld.gov.au; 5Royal Hospital for Women, Randwick, NSW 2031, Australia; 6Faculty of Medicine, University of New South Wales, Sydney, NSW 2033, Australia; 7School of Chemistry & Molecular Biosciences, University of Queensland, Brisbane, QLD 4072, Australia; m.dekker@uq.edu.au

**Keywords:** SPRING, AMPK, microbiome

## Abstract

**Background/Objectives:** Dietary fibre promotes health, partly by mediating gut microbiota and short-chain fatty acid (SCFA) production. Pregnancy alters the relationship between dietary composition and the gut microbiota, and it is unclear if fibre intake during late pregnancy alters the abundance of SCFA bacteria and circulating SFCA concentrations. The aim of this study was to determine the impact of dietary fibre on faecal microbiome composition and circulating concentrations of SCFA acetate, butyrate, and propionate in late pregnancy. We also aimed to assess the impact of propionate treatment on placental function using cultured placental explants. **Methods:** 16S rRNA gene amplicon sequencing was performed on faecal DNA collected at 28 weeks of gestation from participants enrolled in the SPRING cohort study consuming a low or adequate fibre diet. Circualting SCFA were assessed. Placental explants were treated with sodium propionate. **Results**: Fibre intake did not impact microbial diversity or richness but did impact the abundance of specific bacterial genera. Pregnant participants with low-fibre diets had a greater abundance of *Bacteroides* and *Sutterella*, and dietary fibre intake (mg/day) negatively correlated with genera, including *Sutterella*, *Bilophila*, and *Bacteroides*. SCFA concentrations did not differ between groups but circulating concentrations of acetate, propionate, and butyrate did correlate with the abundance of key bacterial genera. Propionate treatment of placental explants did not alter mRNA expression of fatty acid receptors, antioxidants, or markers of apoptosis, nor did it impact pAMPK levels. **Conclusions**: This study demonstrates that the impact of dietary fibre on SCFA concentrations in pregnant women is modest, although this relationship may be difficult to discern given that other dietary factors differed between groups. Furthermore, this study demonstrates that propionate does not impact key pathways in placental tissue, suggesting that previous associations between this SCFA and placental dysfunction may be due to other maternal factors.

## 1. Introduction

Poor nutrition associated with the Western diet has been linked to the rise in obesity and non-communicable diseases observed across developed nations [1]. This extends to pregnancy, with poor dietary habits and excess weight linked to suboptimal maternal environment, pregnancy complications, and adverse foetal outcomes [2]. These factors can contribute to long-term deficits in offspring health, leading to an intergenerational cycle of poor health and non-communicable disease [3]. Traditionally, dietary components are considered in terms of their effects on systemic homeostasis once absorbed into circulation. However, diet also contributes to health through its influence on the composition of the gut microbiota [4]. Dietary fibre is of particular interest in this context as it is non-digestible but still impacts overall health through pathways, including but not limited to the regulation of the microbial composition of the gut [5]. Studies investigating the impact of Western diets on pregnancy outcomes often overlook the fact that while macro- and micronutrients are dysregulated, fibre intake is also often severely decreased. We have previously demonstrated in the SPRING study that the composition of the microbiome at 16 weeks of gestation is associated with BMI [6], and that dietary fibre intake negatively correlates with a key bacterial genus that itself is associated with higher insulin concentrations [7]. We also demonstrated that the few overweight and obese pregnant participants who were vegetarian had a different microbiome composition at 16 weeks of gestation compared to those who were omnivorous [8]. The samples for gastrointestinal microbiome analysis were also collected at 28 weeks gestation, and the impact of overall diet quality was assessed [9]; however, the specific impact of dietary fibre intake on outcomes at this later time point was not characterised.

Higher dietary fibre intake has been associated with a greater abundance of “health-promoting” bacteria. This is thought to be driven by the fact that dietary fibre can be fermented by beneficial gut microbiota to generate energy for microbial growth while forming short-chain fatty acids (SCFAs) as a beneficial waste product [4,10]. SCFAs contribute to the maintenance of gut health by impacting both bacterial composition and intestinal physiology. SCFAs bind to free fatty acid receptor 2 (FFAR2) and FFAR3 in colonocytes to induce multiple cell signalling pathway intermediates that regulate metabolic pathways in the mitochondria and cytoplasm [4,10,11]. These SCFAs can also be absorbed into circulation, although they are found at concentrations that are up to 100-fold lower than levels seen in the intestines [12]. However, these low circulating concentrations still are sufficient to impact some of these same metabolic pathways in cells normally found in distant tissues [13]. Many of these pathways are present in the placenta and act to maintain processes required for a healthy pregnancy [14,15,16]. A study by Wang et al. investigated circulating SCFA concentrations and demonstrated that propionate levels were decreased in women with key pregnancy disorders and placental dysfunction [17]. However, it remains unknown if these associations are causative, as no studies have investigated the direct effects of SCFAs on placental tissue. Given the prevalence of poor nutrition and obesity across the world, it is important to understand the role of dietary fibre on gut microbiota modulation and SCFA levels in the context of pregnant overweight and obese women, as well as the effects of SCFAs on the placenta. This study aimed to compare gut microbial composition in pregnant participants who have either adequate fibre intake or low fibre intake during pregnancy. This study also assessed serum SCFA concentrations in these participants and characterised the impacts of SCFA exposure on key signalling pathways in placental explants to investigate a possible mechanism by which maternal dietary fibre intake affects the placenta and, thereby, foetal health.

## 2. Materials and Methods

### 2.1. Study Population

The influence of maternal dietary fibre intake on gut microbiota composition and serum SCFA levels was evaluated using samples collected at 28 weeks gestation from a subset of pregnant participants who were overweight or obese from the SPRING (Study-of Probiotics-in-the-Prevention-of-Gestational Diabetes Mellitus) cohort study (ACTRN12611001208998) [18]. This study was approved by the Royal Brisbane and Women’s Hospital Human Research Ethics Committee (HREC/11/QRBW/467), the University of Queensland Human Research Ethics Committee (UQ2012000080), and the Mater Hospital Human Research Ethics Committee (Mater1894M). All participants provided written informed consent, and the study was conducted in accordance with the Declaration of Helsinki. Habitual dietary intake was assessed at 28 weeks of gestation through the Cancer Council Victoria’s Dietary Questionnaire for Epidemiological Studies (DQESv2). A stool sample was provided by study participants at 28 weeks of gestation. In this sub-study, participants were excluded based on body mass index (BMI) > 45 kg/m^2^, medication use other than non-fibre supplements, the presence of pregnancy complications, and offspring in <10th percentile for birth weight. From the habitual dietary assessment, energy-adjusted dietary fibre intake (mg/kJ) was calculated and used to separate women into low-fibre (1st quartile) and adequate-fibre (4th quartile) diet groups. This resulted in a sample size of *n* = 27 in the low-fibre group and *n* = 25 in the adequate-fibre group. Of these women, *n* = 17 from the low-fibre group and *n* = 23 from the adequate-fibre group had serum available for UPLC-MS/MS analysis at 28 weeks. All samples were matched for maternal and offspring parameters.

### 2.2. Microbiota—Composition

DNA extraction and 16S rRNA gene amplicon sequencing were undertaken on stool samples from the SPRING cohort study, which generated a dataset of the microbiota composition of all participants in the study. The microbiome dataset generated was then used to investigate several key research questions with details of how the samples were collected, extracted and bacterial composition assessed, as described previously [19]. Briefly, DNA was extracted from stool samples collected at 28 weeks of gestation, and 16S rRNA gene amplicon sequencing of the V6–V8 region was conducted using the Illumina MiSeq platform at the University of Queensland Australian Centre for Ecogenomics. The V6–V8 hypervariable regions of bacterial 16S rRNA were PCR amplified using the 926F forward (5′-TCG TCG GCA GCG TCA GAT GTG TAT AAG AGA CAG AAA CTY AAA KGA ATT GRC GG-3′) and 1392R reverse (5′-GTC TCG TGG GCT CGG AGA TGT GTA TAA GAG ACA GAC GGG CGG TGW GTR C-3′) primers. The positive control was *Escherichia coli* (JM109), and the negative control was sterilised deionised water. PCR amplicons were purified using AMPure XP Beads (New England BioLabs, Notting Hill, Australia). The sequencing library was generated using the Nextera XT Index kit (Illumina, Melbourne, Australia), followed by library quantification, normalisation, and pooling as per the manufacturer’s instructions. Downstream processing of sequences was conducted using QIIME (Quantitative Insights Into Microbial Ecology, v1.9.1). The sequences were allocated to operational taxonomic units (OTUs) using the Greengenes reference database when pairwise identity was >97% [20].

To validate microbiota results, qPCR was performed on previously extracted faecal DNA using primer sets detailed previously [6,21,22,23,24], and as shown in Appendix A. All qPCRs used the SYBR Green QuantiNova PCR MasterMix, contained a positive and negative control, and were conducted using the QuantStudio 6 Flex Real-Time PCR System (Applied Biosystems, Thermo Fisher, Brisbane, Australia) using primer sets for *Bilophila*, *Sutterella*, *Odoribacter*, *Faecalibacterium*, and total bacteria (16S rRNA gene). The geomean of each triplicate was determined, and the genera abundance was calculated using the 2^−ΔΔC(t)^ method; the results of the total bacteria primer set were used as the endogenous control.

### 2.3. Serum—Short-Chain Fatty Acids

SCFA concentrations were measured in serum collected at 28 weeks using Ultra-Performance Liquid Chromatography–Tandem Mass Spectrometry (UPLC-MS/MS), as previously described [25]. Briefly, SCFA were extracted from 10 µL of the serum using 100 µL of 5 µM 2-Ethylbutyric Acid (109959, Sigma-Aldrich, Melbourne, Australia) in 100% Acetonitrile (ACN, 34851, Sigma-Aldrich, Melbourne, Australia). Samples were derivatised in 45 µL of 200 mM 3-nitrophenylhydrazine (30% ACN) and 45 µL of 120 mM *N*-(3-dimethylaminoproyl)-N1-ethylcarbodiimide solution (6% Pyridine, 100% ACN). The samples were incubated for 30 min at 40 °C before the addition of 1.8 mL of 10% ACN and stored at 4 °C. UPLC-MS/MS was performed using the Nexera UPLC series system (Shimadzu, Brisbane, Australia) and the SCIEX5500 QTRAP mass spectrometer (AB-SCIEX Instruments). The peak area was determined using the MultiQuant Software 3.0 (SCIEX, Mt Waverley, Australia). Serum SCFA concentrations were calculated from a standard curve generated using the Volatile Fatty Acid Mix (CRM46974, Supelco, Sigma-Aldrich, Melbourne, Australia) and glacial acetic acid (A6283, Sigma-Aldrich, Melbourne, Australia) ranging from 0.5 to 250 µM.

### 2.4. Short-Chain-Fatty-Acid Treatment of Placental Explants

Term placentae were collected from the Mater Mother’s Hospital (Brisbane, QLD, Australia) following planned caesarean section. Participants were recruited to the *Peripartum-Maternal-Microbiome:-Relation-to-Gestational-Diabetes-and-Metformin-Use* study. Ethics was provided by the Mater Misericordiae Ltd. Human Research Ethics Committee (HREC/MML/53814) and the University of Queensland (2019/HE003099). All participants provided written informed consent, and the study was conducted in accordance with the Declaration of Helsinki.

Placental tissue was collected as per the Global Pregnancy Collaboration protocol [26]. Five tissue biopsies were collected from each placenta before the removal of maternal membranes and decidua tissue. The remaining tissue was then placed in CMRL-1066 culture media (11530037, ThermoFisher Scientific, Brisbane, Australia) containing a 5% foetal Bovine Serum, 1X Antibiotic–Antimycotic, 10 mM HEPES, 2.5 mM L-Glutamine, and 0.1 ug/mL retinyl-acetate (CMRL-1066^+^). These biopsies were further dissected into ~2 mm^3^ segments and combined to produce a homogenous mixture that was representative of the entire placenta and likely included a variety of cell types. In total, 120 ± 15 mg of wet-weight placental tissue was transferred to individual wells of a plastic 24-well culture plate containing 1 mL of CMRL-1066^+^ media. Tissue explants were left overnight to equilibrate with agitation at 37 °C with 5% CO_2_. Media were then replaced with CMRL-1066^+^ containing treatments.

Previous studies have treated cells with doses of propionate that are well above the concentration seen in plasma, with concentrations ranging from 100 µM up to 20 mM being commonly used [27,28,29]. While concentrations of propionate are high within the lumen of the gut (over 140 µM), they are commonly reported in the range of 1–10 µM in serum or plasma [12]. Given that we have measured circulating propionate concentrations to be approximately 1 µM in participants of the current study, we selected doses of propionate that were close to the physiological range. As such, the CMRL-1066^+^ was supplemented with a vehicle (ultra-pure H_2_O) of 7 µM or 20 µM of sodium propionate (P5436, Sigma-Aldrich, Melbourne, Australia) for 4 or 24 h. The explants were collected, snap-frozen in liquid nitrogen, and stored at −80 °C for later use. Biopsies from four placentae were used in the present study, with 3 technical replicates per treatment per placenta (see Appendix A for participant summary).

### 2.5. Mitochondrial DNA Content and Gene Expression

DNA was extracted from treated explant tissue via isopropanol precipitation. qPCR was performed using two primer sets that amplify mitochondrial DN, and two others that amplify nuclear DNA (Appendix A) to determine the ratio of normalised mitochondrial DNA to nuclear DNA, as described previously [16]. RNA was extracted from explant tissue using the RNeasy Mini Kit (74106, Qiagen, Clayton, Australia), reverse transcribed into cDNA, and qPCR was performed using KiCqStart SYBR Green Primers (Appendix A), as previously described [30].

### 2.6. Protein—Extraction and Western Blotting

Total protein was extracted from explanted placental tissue at the 4 hr time point. Tissue was homogenised with 300 µL from the RIPA buffer containing protease (Sigma-Aldrich, Melbourne, Australia) and phosphatase (Roche, North Ryde, Australia) inhibitors and protein concentration diluted to 1 mg/mL in the RIPA buffer for Western blotting. In total, 12 µg of protein per sample were subjected to SDS-PAGE on 12% gels, transferred to a Immun-Blot PVDF membrane (1620177, BioRad, Granville, Australia), and blocked with the Odyssey TBS blocking buffer (LI-COR, Millennium Science, Mulgrave, Australia). Membranes were incubated overnight in antibodies targeting phospho-AMPKα (Thr172) (1:500, *Cat #2535*, Cell Signalling Technology, Notting Hill, Australia), total AMPKα (1:500, *Cat #5831*, Cell Signalling Technology, Notting Hill, Australia), or α-tubulin (1:1000, *Cat #Ab7291*, Abcam, Melbourne, Australia) at 4 °C before the addition of secondary antibodies. Fluorescence was quantified using ImageStudio-Lite (LI-COR, Millennium Science, Mulgrave, Australia) and protein levels were normalised to α-tubulin.

### 2.7. Statistical Analysis

The microbiota data of the 52 participants assessed in the current study were used to investigate the relationship between fibre intake and bacterial composition via the Calypso 8.72 web server [31], which is no longer available for public use. Microbial richness and diversity within groups (α-diversity) were determined using the Chao1 Index and the Shannon Index. Microbial diversity between groups (β-diversity) was measured using an unsupervised Principal Coordinate Analysis (PCoA) and the supervised Canonical Correspondence Analysis (CCA). Analysis of similarities (ANOSIM) was used to compare variances between low- and adequate-fibre diet groups. Differences in general abundance between groups were assessed using the Wilcoxon-rank test. All subsequent analyses were performed using GraphPad Prism 8.4 (GraphPad-Software, Boston, MA, USA). The normality of all data was determined through the Shapiro–Wilk normality test before further analysis. Data comparing the two dietary groups were assessed using the two-tailed unpaired *t*-test to see if they were normally distributed, or a two-tailed Mann–Whitney U test to see if they were not normally distributed. Fisher’s exact test was used for categorical data. Spearman correlations were performed to determine relationships between bacterial genera and fibre, as well as nutrient intake and SCFA concentrations. Explant data were analysed using one-way analysis of variance (ANOVA). Significance was accepted when *p* < 0.05.

## 3. Results

Maternal age was the only clinical characteristic to differ between diet groups and was lower in the low-fibre diet group compared to the adequate-fibre diet group (Table 1). Maternal fasting glucose, C-Peptide, insulin, cholesterol, triglycerides, HDL, LDL and VLDL were not different between fibre groups. Offspring sex, gestational age, birth weight, and mode of delivery were similar between groups (Table 1).

The low-fibre diet group had both lower energy-adjusted fibre (*p* < 0.0001) and gross dietary fibre (*p* < 0.0001) intake when compared to the adequate-fibre group (Table 2). Total energy intake (kJ/day) was similar between groups. Relative to women with adequate-fibre diets, women with low-fibre diets had lower gross- and energy-adjusted carbohydrates (*p* < 0.05) and starch intake (*p* < 0.05). The low-fibre diet group also had lower energy-adjusted fat (*p* < 0.001) and polyunsaturated fat (*p* < 0.05) intake, but a higher energy-adjusted saturated (*p* < 0.0001) and monounsaturated fat (*p* < 0.05) intake when compared to the adequate-fibre diet group. No other differences in macronutrient intake were present (*p* > 0.05, Table 2).

Gross macronutrient intake (g/day) was used to calculate the percentage of kilojoules sourced from fat, proteins, and carbohydrates to allow for a comparison with current dietary guidelines: (45–65%) for carbohydrates, 15–25% for protein, and 20–35% for fats [32]). Both groups consumed similar proportions of protein within the recommended range (Table 2). On average, the low-fibre diet group sourced less energy from carbohydrates when compared to the adequate-fibre diet group (*p* < 0.0001), and both groups consumed fewer carbohydrates than the recommended reference range. Conversely, the low-fibre diet group sourced more energy from fats relative to the adequate-fibre diet group (*p* = 0.0018), and both groups consumed more than the recommended reference range.

### 3.1. Gut Microbiota Composition

Maternal dietary fibre was not associated with α-diversity, with microbial diversity (Figure 1A, Shannon Index, *p* = 0.44) and microbial richness (Figure 1B, Chao1 Index, *p* = 0.1) remaining similar between low- and adequate-fibre diet groups. Microbial diversity between the low- and adequate-fibre groups (β-diversity) was also the same independent of whether supervised (CCA, Figure 1C, *p* = 0.30) or non-supervised (PCoA, Figure 1D, *p* = 0.32) or hierarchical clustering analysis or analysis of similarity (ANOSIM, Figure 1E, r = 0.03, *p* = 0.09) was used.

### 3.2. Dietary Fibre Intake and Genera Abundance

The linear discriminant analysis (LDA) of effect size (LEfSe) showed that the low-fibre diet group had a significantly lower abundance of *unclassified Clostridiales* and a significantly higher abundance of *Sutterella* relative to the adequate-fibre diet group (Figure 1F, LDA > 3.0). Pregnant participants with low-fibre diets had a significantly greater abundance of *Bacteroides* (*p* = 0.03) and *Sutterella* (*p* = 0.04) when compared to the participants with adequate-fibre diets (Figure 1G). *Bilophila* (*p* = 0.06), *unclassified Enterobacteriaceae* (*p* = 0.06), *unclassified Clostridiales* (*p* = 0.07), and *Faecalibacterium* (*p* = 0.09) all appeared to be impacted by dietary fibre, but this impact did not reach statistical significance.

### 3.3. Macronutrient Intake and Genera Abundance

Dietary fibre intake negatively correlated with the abundance of *Sutterella* (r = −0.34, *p* = 0.01), *Bilophila* (r = −0.32, *p* = 0.02), and *Bacteroides* (r = −0.3, *p* = 0.03, Figure 2). *Sutterella* abundance also correlated with carbohydrates (r = −0.4, *p* = 0.004), fat (r = 0.38, *p* = 0.005), saturated fat (r = 0.38, *p* = 0.002), and monounsaturated fat (r = 0.38, *p* = 0.006) intake. Additionally, the abundance of *Faecalibacterium* positively correlated with carbohydrate (r = 0.35, *p* = 0.01) and starch (r = 0.42, *p* = 0.002) intake. Monounsaturated fat intake positively correlated with *Bilophila* (r = 0.3, *p* = 0.03) and *Odoribacter* (r = 0.33, *p* = 0.02) abundance. No significant correlations were observed between genera abundance and maternal polyunsaturated fat intake.

### 3.4. qPCR Validation

There was a positive correlation between the bacterial abundance of *Bilophila, Sutterella,* and *Odoribacter*, as determined by the qPCR amplification with bacterial abundance, as determined by 16S sequencing (Appendix A). qPCR data for *Faecalibacterium* was the only genus that did not significantly correlate with 16S rRNA reads (*r* = −0.1166, *p* = 0.42).

### 3.5. Dietary Fibre and Circulating Short-Chain Fatty Acid Levels

As dietary fibre is a substrate for SCFA production by various bacterial species, maternal serum SCFA concentrations were measured in pregnant women at 28 weeks of gestation. The concentration of circulating SCFAs did not significantly differ between groups, with acetate (Figure 3A, *p* = 0.35), propionate (Figure 3B, *p* = 0.41), and butyrate (Figure 3C, *p* = 0.35) concentrations remaining similar despite differential dietary-fibre intake.

To assess whether habitual macronutrient intake (mg/kJ) was associated with circulating SCFA levels, dietary data were correlated with circulating SCFA concentrations (Table 3). Serum butyrate concentrations correlated positively with energy-adjusted carbohydrate (*r* = 0.35, *p* = 0.03), but negatively with saturated fat (*r* = −0.38, *p* = 0.01), and monounsaturated fat (*r* = −0.32, *p* = 0.04) intake. No other correlations were identified between dietary components and circulating SCFA concentrations.

### 3.6. Circulating Short-Chain Fatty Acids and Genera Abundance

Correlation analysis of genera and circulating SCFA levels independent of dietary fibre intake was then investigated (Figure 3D). The SCFA-producer *Faecalibacterium* was positively correlated with serum acetate (*r* = 0.44, *p =* 0.005), propionate (*r =* 0.4, *p* = 0.01), and butyrate (*r* = 0.36, *p =* 0.02). *Odoribacter* abundance was negatively associated with both serum propionate (*r* = −0.27, *p* = 0.04) and butyrate (*r* = −0.42, *p* = 0.007). None of the remaining genera previously observed to be related to maternal dietary fibre, including *Bacteroides*, *Bilophila*, *Sutterella*, *Unclassified Clostridiales*, and *Unclassified Enterobacteriaceae,* were significantly correlated with circulating maternal SCFA concentrations.

### 3.7. Impact of Short-Chain Fatty Acids on Placental Explants

To investigate the effects of SCFAs on placental tissue, placentae were collected from four overweight or obese women without any complications of pregnancy following a planned caesarean section (see Appendix A). Given that serum propionate concentrations correlated with five different bacterial genera in pregnant women, that several of the genera of interest impacted by fibre intake were propionate-producers, and that links between serum propionate and gestational disorders implicate placental dysfunction [17], we selected propionate to investigate its effects on placental explants.

Propionate treatment did not impact the expression of *FFAR2* (Figure 4A) at 4 h (*p* = 0.47) or 24 h (*p* = 0.27). Similarly, propionate treatment had no effect on *FFAR3* mRNA levels (Figure 4A) at 4 h (*p* = 0.25) or 24 h (*p* = 0.33). Propionate had no effect on the mRNA expression of antioxidant genes (Figure 4B) *CAT* at 4 h (*p* = 0.89) or 24 h (*p* = 0.16), *SOD1* at 4 h (*p* = 0.87) or 24 h (*p* = 0.94), or *SOD2* at 4 h (*p* = 0.09) or 24 h (*p* = 0.91). Propionate had no effects on apoptotic gene mRNA expression (Figure 4C), with no differences in *CASP3* at 4 h (*p* = 0.60) or 24 h (*p* = 0.44) or *CASP8* at 4 h (*p* = 0.92) or 24 h (*p* = 0.57).

SCFAs have also been shown to stimulate mitochondrial biogenesis. As such, we assessed mitochondrial content. Mitochondrial DNA content (Figure 4D) did not differ between groups following propionate exposure for 4 h (*p* = 0.74) or 24 h (*p* = 0.60).

As SCFAs have been shown to activate AMPK-signalling in other tissues, phosphorylated AMPKα and total AMPKα protein levels were examined in placental tissue treated with propionate for 4 h (Figure 4). Phosphorylated AMPKα (Figure 4E, *p* = 0.61) and total AMPKα (Figure 4F, *p* = 0.2) levels and the ratio of phosphorylated AMPKα to total AMPKα (Figure 4G, *p* = 0.75) were all not impacted by propionate exposure.

## 4. Discussion

In this study, we compared faecal microbial composition and circulating SCFA concentrations at 28 weeks of gestation in overweight and obese pregnant participants who consumed either a low- or adequate-fibre diet. Here, we show that maternal dietary fibre intake did not impact microbial diversity or richness, but did impact the abundance of specific bacterial genera. Surprisingly, dietary fibre did not impact the abundance of healthy SCFA-producing bacteria or circulating SCFA concentrations. Given that serum propionate concentrations correlated with several bacterial genera in pregnant women, and that the links between serum propionate and gestational disorders implicate placental dysfunction [17], we next treated placental explants with propionate to determine if it directly impacted key signalling pathways. Despite the key receptors being present in placental tissue, propionate did not impact key pathways known to be impacted by SCFAs in other tissues. This study suggests that any potential benefits of fibre intake on pregnancy outcomes occur independent of changes to SCFA-producing bacteria and that physiological concentrations of propionate have no impact on term placental physiology.

The impact of diet on microbial diversity and richness in pregnant individuals has previously been investigated, but few studies have characterised the impact of dietary fibre on the gastrointestinal microbiome at 28 weeks of gestation. Outside pregnancy, multiple studies have shown that fibre intake impacts the community structure of the gut microbiome (reviewed by [33]). Here, we demonstrate that markers of community structure, including α-diversity and β-diversity, were not different between adequate and low-fibre diet groups. In our previous assessment of the microbiome at 16 weeks of gestation, we similarly demonstrated that overall α and β diversity did not differ between participants who had consumed low- and high-fibre diets [7]. In contrast, a Finnish study reported that fibre intake, when not corrected for energy intake, was correlated with both microbial diversity and richness in samples collected from pregnant women before the 17th week of gestation [34]. That study differs from the current study in that they investigated outcomes at the operational taxonomic level rather than the genus level, which may partially explain differences in outcomes. The Finnish study also reported that a similar relationship was also present for total carbohydrate intake [34]. Furthermore, they also demonstrated that dietary fats as a percentage of total energy intake had the opposite effects on community structure to fibre. Our current study similarly demonstrated that separating participants into groups based on fibre intake coincides with changes to both carbohydrate and fat intake. While we did not undertake additional analyses to assess the impact of these other dietary factors on overall bacterial diversity, it is likely that changes to these other dietary factors may have influenced our overall analysis. This is a limitation of the current study.

A key finding from this study was that despite there being no changes in the community structure, the abundance of several specific bacterial genera differed between fibre groups, with a greater abundance of *Sutterella* and *Bacteroides* in low-fibre-consuming women at 28 weeks of gestation. We similarly demonstrated that low fibre intake increased *Sutterella* abundance in this cohort of women at 16 weeks of gestation [7]. *Bacteroides*, *Bilophila*, and *Sutterella* abundance were negatively correlated with energy-adjusted fibre intake. In fact, energy-adjusted fibre intake was the only dietary factor that was shown to correlate with *Bacteroides* abundance. As such, despite multiple dietary factors differing between groups, it is likely that the difference in the abundance of *Bacteroides* between low- and adequate-fibre groups was indeed due to differences in fibre. *Bacteroides* abundance outside of pregnancy is associated with Western dietary patterns, with lower abundances linked to high-fibre and plant-based diets [35,36]. The elevated *Bacteroides* abundance in the low-fibre diet group is potentially harmful, as high *Bacteroides* abundance is generally thought to be pro-inflammatory and a risk factor for the development of metabolic disease [37,38]. This has been corroborated in the context of pregnancy, with greater *Bacteroides* abundance reported in overweight and obese individuals compared to healthy-weight individuals [39]. Additional positive correlations between *Bacteroides* abundance and excessive gestational weight gain, HbA1C, CRP, and LPS biosynthesis pathways have also been observed across several studies [39,40,41]. In contrast, the other genus that was lower in the adequate-fibre group compared to the low-fibre group was *Sutterella*, and its abundance was negatively correlated with fibre intake but was also negatively correlated with total carbohydrates and positively correlated with total fat, saturated fat, and monounsaturated fat intake. This is perhaps unsurprising as dietary fibre is commonly found in carbohydrate-rich foods, and individuals with a lower carbohydrate diet will likely consume lower fibre and higher fat content. As such, we cannot rule out the possibility that fat and carbohydrate intake may have had a major contribution to the difference in this bacterial genus between groups. Indeed, greater *Sutterella* abundance has been linked to Western dietary patterns low in fibre and high in animal products, a combination that may favour *Sutterella* expansion [42]. *Sutterella* is bile-tolerant and, therefore, resistant to the higher bile levels associated with the consumption of higher fat content [43].

Importantly, qPCR successfully validated the 16S sequencing data for three of the four bacterial genera assessed. While qPCR amplification of the sequence detected by the *Faecalibacterium* primers did not correlate with the 16S rRNA data, we would like to highlight that this is likely due to the primers selected for this validation. There are many species of *Faecalibacterium* and our primers are very specific for just one species of *Faecalibacterium*, *F. duncaniae*. This species is also not the most common *Faecalibacterium* species, and as such, changes to this one species are not necessarily representative of the entire genera. Subsequent studies should validate 16S sequencing using primers that detect sequences common to multiple bacterial species within that genus.

We have previously reported findings from a larger sub-group of the SPRING cohort demonstrating that the bacterial genus *Collinsella* was associated with insulin concentrations and fibre intake at 16 weeks of gestation [6,7,8]. Interestingly, by 28 weeks of gestation in this specific and small cohort, there was no correlation between fibre intake *and Collinsella* abundance, nor was it differentially abundant between the groups. It is likely that as pregnancy advances and placental hormones start to shift metabolic processes, the relationship between diet, bacterial abundance, and metabolic outcomes also changes. We have shown this to be true even in rats with carefully controlled diets and environmental settings [44], highlighting that studies such as the current student are required to ascertain if well-established relationships between diet and microbiome remain in late pregnancy.

Results from the current study were not in line with the hypothesis that more fibre consumption would translate to a greater abundance of ‘health-promoting’, SCFA-producing bacteria at 28 weeks of gestation. This may be because, within the cohort, the highest fibre-consuming individuals were still only consuming fibre at the recommended levels as opposed to very high levels. Indeed, the adequate-quality diet group did not have a greater abundance of genera typically associated with higher fibre intake. Based on these findings, it is unsurprising that we found no difference in serum acetate, propionate, or butyrate levels between women consuming low- versus adequate-fibre diets at 28 weeks of gestation. Based on such minimal effects of fibre on the abundance of SCFA-producing bacteria and no change in SCFAs in plasma, we did not measure SCFAs in faecal material. If we had assessed faecal SCFA content and found it also unaffected by fibre, this would have further supported our findings that diet has a lesser effect on these bacterial genera compared to pregnancy itself. Despite not measuring faecal SCFAs, we did find that the abundance of one bacteria genus was associated with plasma SCFA levels. *Faecalibacterium* of the Clostridia class positively correlated with all three circulating SCFAs. *Faecalibacterium* is a well-known acetate-consuming, butyrate-producing bacterium. Whilst *Faecalibacterium* abundance did not correlate with measures of dietary fibre consumption, it was positively associated with energy-adjusted carbohydrate and starch intake, which have been linked to *Faecalibacterium* expansion [45]. Whilst the present study cannot rule out differences in SCFA production in the large intestine, these findings possibly indicate that serum SCFA concentrations may not be substantially altered by fibre intake at 28 weeks of gestation.

While serum SCFAs were not different between fibre groups, plasma concentrations of SCFAs have been linked to adverse pregnancy outcomes. Indeed, we have measured SCFA levels in a separate subset of participants from the SPRING study who developed late-onset preeclampsia compared to matched controls. We found that the abundance of butyrate-producing Coprococcus  and serum concentrations of butyrate were lower in participants with preeclampsia compared to controls [25]. In the current study, we excluded participants with known pregnancy complications, such as gestational diabetes or preeclampsia, from analysis to avoid any role of those complications driving changes in microbiome composition. Therefore, it is possible that we have removed the individuals that may have driven the previously identified associations between SCFA concentrations, SCFA-producing bacteria, and poor pregnancy outcomes. A study by Wang et al. demonstrated that propionate levels were lower in the second trimester of women with GDM and that propionate was positively correlated with insulin levels in both the second and third trimester [17]. That same study demonstrated that mRNA expression of key SFCA receptors, FFAR2 and FFAR3, were decreased in placental tissue from women with GDM. Furthermore, metabolomic profiling of that placental tissue demonstrated disrupted ATP generation in placental tissue. Given that propionate can impact ATP/ADP ratios in other tissues, it is possible that this change in placental tissue in GDM pregnancies was related to the propionate concentrations. However, it is also possible that changes in maternal glucose or insulin in GDM participants directly impact these pathways in placental tissue. In another study by Hu et al., faecal propionate levels were negatively correlated with placental protein carbonyl levels [46]. They also found that placental protein carbonyl levels correlated with serum progesterone concentrations, which once again makes it difficult to assign causality between propionate and placental dysfunction. In an attempt to remove alternative maternal influences on placental outcomes, a mouse study was conducted in which a propionate-rich diet was reported to prevent placental insufficiency and restore foetal weight in a mouse model of foetal growth restriction [47]. However, it remains uncertain if propionate improved outcomes by modifying maternal health or directly via its actions on the placenta. This highlighted the need to assess if propionate directly impacted key signalling pathways in placental tissue.

As such, we conducted placental explant cultures to assess if propionate impacted key signalling pathways in placental tissue. Given that we wanted to assess if a propionate treatment at the concentrations seen in the plasma impacts the placental outcomes, we exposed placental explants to sodium propionate at 7 µM or 20 µM compared to the vehicle alone. We demonstrated that these physiological concentrations of propionate for 4 or 24 h did not change mRNA expression of FFAR2 or FFAR3 nor of antioxidant enzymes or regulators of antioxidants. We also demonstrated that propionate did not impact mitochondrial content or levels of phosphorylated AMPKα. As such, propionate treatment at the doses and durations examined in the current study had no impact on any of the placental parameters assessed. While there is a large body of evidence demonstrating that the abundance of SCFA-producing bacteria correlates with plasma SCFA levels and overall health [48], there is less evidence for a direct role of these SCFAs on peripheral tissues. Dietary supplementation of propionate can increase plasma propionate levels and improve overall health and Cell culture experiments have shown the direct actions of propionate on cell function. Shimizu et al. demonstrated in rodents that dietary supplementation with 5% propionate in food increases plasma propionate from about 7 µM up to about 12 µM, and this magnitude of change was associated with drastic improvements in metabolic function [49]. However, studies that have investigated how these SCFAs impact cellular outcomes in culture, typically use doses that are much higher. Wang at et demonstrated, using the colorectal cell line Caco-2, that the concentration of propionate required to increase the Phosphorylated AMPK to total AMPK ratio is 500 µM with 100 µM not having any effect [50]. Another study demonstrated that hepatocytes cultured in 2000 µM of propionate, a dose 100-fold higher than seen in plasma in humans, display improved antioxidant activity with both SOD and glutathione peroxidase being increased, but they did not investigate outcomes using lower concentrations [51]. This same study also demonstrated that propionate induces its effects in a time-dependent manner with increases in hepatocyte expression of metabolic genes greater at 24 h than at 12, 6 or 3 h. However, they did demonstrate that key effects had started to occur by 3 h of exposure [51], which highlights that while the doses used in the current study might not be high enough to induce an effect, we have investigated outcomes at relevant time points. It is quite possible that the concentrations of propionate seen by the colon and liver are much higher than the doses seen in circulation, so the concentrations used in the studies that investigate the effects of propionate on gastrointestinal health are likely physiologically appropriate. However, a range of studies have also investigated the effects of propionate in tissues, which would more likely be exposed to doses seen in circulation. Given the clinical associations between gut microbial communities and heart and kidney health, studies have investigated the potential benefit of propionate exposure on cardiomyocytes or proximal tube cells. Cardiomyocytes cultured in 10 µM and 1000 µM were found to have mitochondrial respiration similar to untreated cells [52]. In contrast, treating perfused rat hearts with 3000 µM propionate induced a number of changes to key heart metabolites and enhanced metabolic flux from glucose to acetyl-CoA [53]. Another study treated HK2 cells (human proximal tubule kidney cell line) with 15,000 µM propionate and demonstrated changes in gene expression that were markedly greater than the effects of the inflammatory agent TNF-α [54]. Collectively, these studies highlight that while propionate does induce a range of effects in various cell populations, the doses required to do so are likely to be much higher than are obtained in circulation by changes to microbial composition alone. Few studies to date have assessed the impact of propionate on placental tissues. Jin et al. treated HTR-8/SVneo cells, a first-trimester trophoblast cell line, with LPS to induce inflammation and then exposed these cells to increasing concentrations of propionate. They demonstrated that 100 µM propionate did not impact key signalling pathways but that doses over 200 µM did prevent the changes to placental signalling pathways that were disrupted by inflammation [55]. While that study did not investigate the expression of antioxidants, mitochondrial number or pAMPK, the studies mentioned above from other tissues have demonstrated that propionate at doses likely seen by the placenta does not impact these pathways. This suggests that the clinical relationship between GDM and placental deficits reported by others may not be directly caused by propionate acting on placental tissue at physiological concentrations.

While this study has advanced our understanding of how fibre intake impacts bacterial abundance and SCFA concentrations in late pregnancy as well as the impact of physiological concentrations of SCFAs on placental outcomes, it is not without its limitations. As described throughout, grouping participants based on fibre intake resulted in multiple dietary factors differing between groups. This is likely related to overall dietary patterns, with higher-fibre foods often containing higher carbohydrate content and lower fat content. Clinically, this is also important as advising patients to increase dietary fibre will likely result in changes to multiple dietary components, and collectively, these changes might influence the microbial composition of the gut.

Another limitation of this study was that bacterial abundance was only assessed in individuals who were overweight or obese. It is well known that the gut microbiome is affected not only by our diet but also by our hormonal milieu and body composition. By selecting overweight and obese individuals, we have likely reduced the variability in microbial composition but have not provided information relevant to individuals who have a lower BMI. As it is, we found SCFA levels to vary significantly within our cohort and broadening the BMI of participants may have increased that variability further. Circulating SCFA levels are likely to change over time so only measuring concentrations at one point in the current study is another limitation of this project. We also relied on self-reported dietary information to group individuals into low- and adequate-fibre groups. There are well-known limitations to using self-reported dietary information to determine the dietary information of study participants. While the DQESv2 has been shown to demonstrate good reproducibility [56], it does rely on recall, and that may impact the quality of the data included. Another challenge related to this study is the fact that the Calypso web server is no longer available. As such, the data from the current paper may not be directly comparable to future studies which may use a different assessment tool; however, the underlying statistical methodology is identical to other available assessment tools.

There are also several future directions for this project. While 16S rRNA gene amplicon sequencing is highly suitable to assess bacterial abundance, it does not inform us of the functional capacity of the gut microbiome. Metagenomics would be a suitable future approach to inform this function. This could be coupled with a more direct assessment of the bacterial SCFA levels by measuring SCFA levels in faecal material. While we chose to only measure circulating SCFAs in the current study, the assessment of foetal SCFA would better inform us about the impact of fibre on SCFA-producing bacteria in pregnant women, though it would by itself not provide insights into their potential effects on host physiology apart from the colon. Finally, we only assessed the impact of propionate treatment on placental outcomes. In future studies, it would be ideal to also assess the impact of butyrate and acetate to broaden our understanding of the effects of SCFAs on placental tissue.

## 5. Conclusions

Overall, the present study has highlighted that low-fibre diets are associated with changes to the microbiome at 28 weeks of gestation. However, adequate fibre intake was not associated with a greater abundance of dietary-fibre-associated genera or higher circulating SCFA concentrations. This suggests that the relationship between dietary fibre consumption, SCFA-producing bacteria, and serum SCFA levels may not be present in late pregnancy. Propionate treatment at physiological concentrations did not impact placental pathways, which might be associated with improving pregnancy outcomes. Based on these results, it is likely that while improving dietary fibre in pregnancy may lead to improved pregnancy outcomes, these improvements may not act through changes to SCFA-producing bacteria or SCFA effects on placental tissue.

## Figures and Tables

**Figure 1 nutrients-17-01234-f001:**
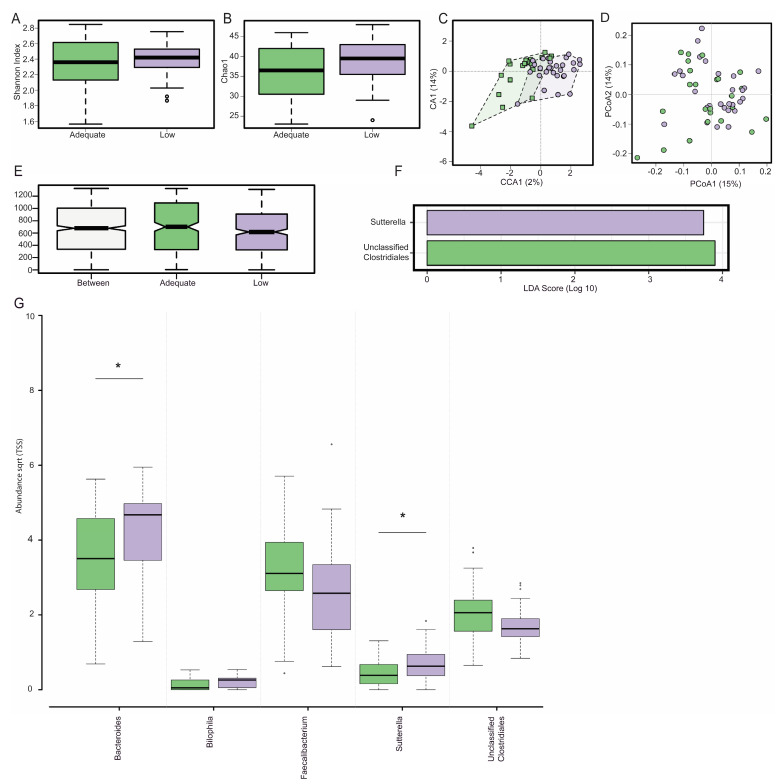
Gut microbiota composition at 28 weeks gestation in pregnant participants with adequate (green) and low (purple) fibre diets, as defined by dietary fibre intake (mg/kJ). There was no effect of dietary fibre on α-diversity as estimated by the Shannon Index (**A**) and the Chao1 Index (**B**) nor was there any effect on β-diversity as estimated by a supervised canonical co-ordinate analysis (**C**) or the unsupervised principal co-ordinate analysis (PCoA) using Bray–Curtis metrics (**D**). The Analysis of similarity using the Bray-Curtis dissimilarity matrix (ANOSIM) depicts intra- or inter-group differences in the gut microbiota composition between groups (**E**). The linear discriminant analysis (LDA) of the effect size (LEfSe) plot demonstrates that *Sutterella* and *unclassified Clostridiales* are the genera most likely to explain differences between adequate- and low-fibre diet groups (**F**). Participants with a low-fibre diet had an increased abundance of *Bacteroides* and *Sutterella* (**G**). Box plots depict a median with the twenty-fifth and seventy-fifth quartiles and the dotted error bars showing the centiles of 2.5 and 97.5, with the dots outside of this depicting outliers. * denotes significance *p* < 0.05. For the LEfSe, significance was accepted when LDA > 3.

**Figure 2 nutrients-17-01234-f002:**
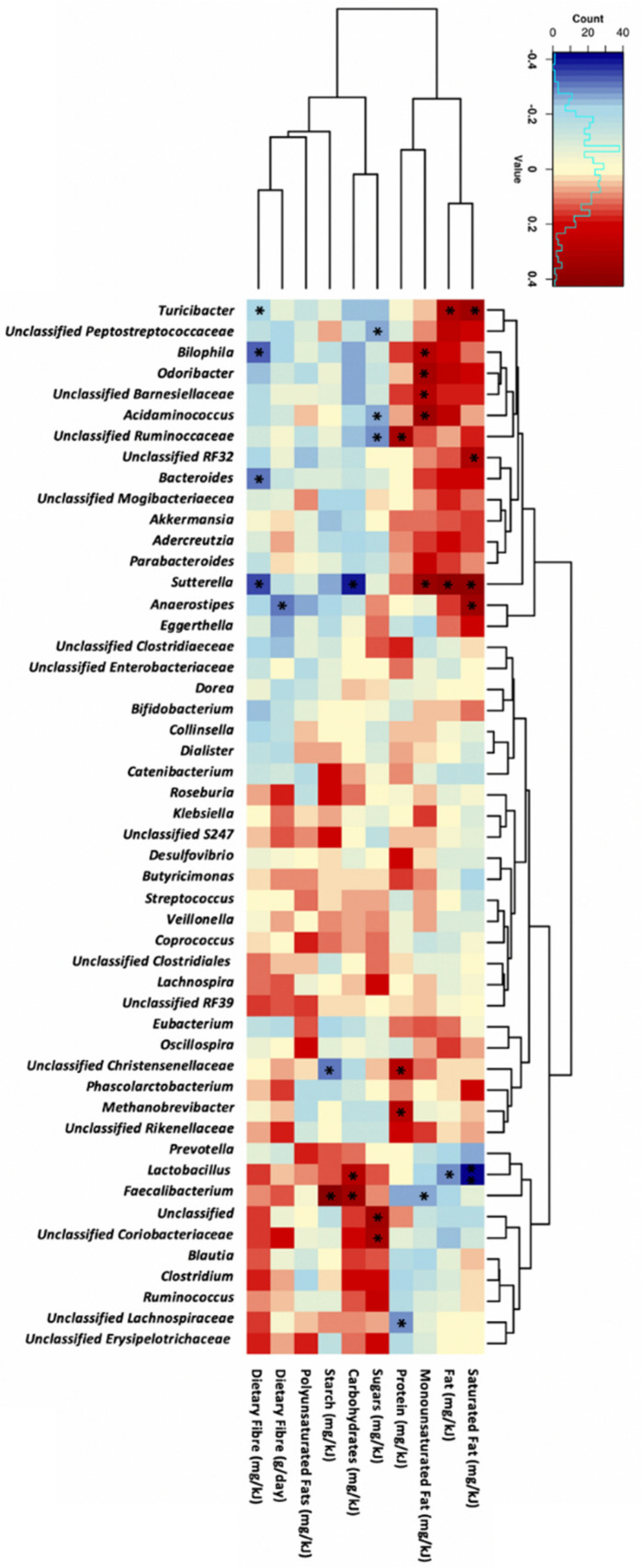
Correlation analysis between the top 50 most abundant genera and maternal dietary intake at 28 weeks of gestation in all participants (*n* = 52). Statistical analysis was performed using Spearman’s correlation, with Spearman’s rank correlation coefficients and *p*-values shown. The darker the red colour, the stronger the positive correlation whereas, the darker blue colour intakes a stronger negative correlation. Significance was accepted when *p* < 0.05, * for *p* < 0.05 and blank for non-significant *p*-values.

**Figure 3 nutrients-17-01234-f003:**
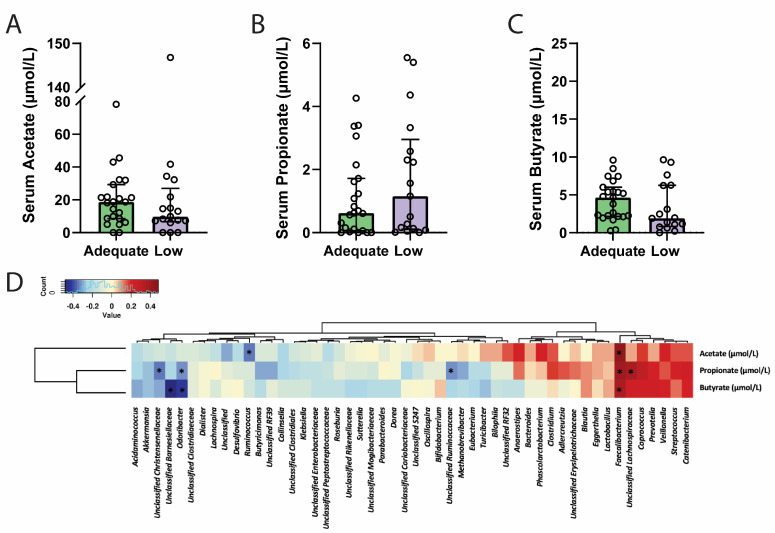
There were no differences in serum acetate (**A**), propionate (**B**) or butyrate (**C**) concentrations between participants from the adequate- and low-fibre groups. Serum acetate, propionate, and butyrate concentrations were found to correlate with the abundance of at least one bacteria genus each, with propionate levels correlating with the abundance of five separate bacterial genera (**D**). Serum concentrations of SCFAs are presented as a median and interquartile range, with individual samples represented by circles in panels (**A**–**C**). * denotes significance *p* < 0.05. Data were analysed using a two-tailed Mann–Whitney U test following Shapiro–Wilk normality testing or Spearman correlation analysis.

**Figure 4 nutrients-17-01234-f004:**
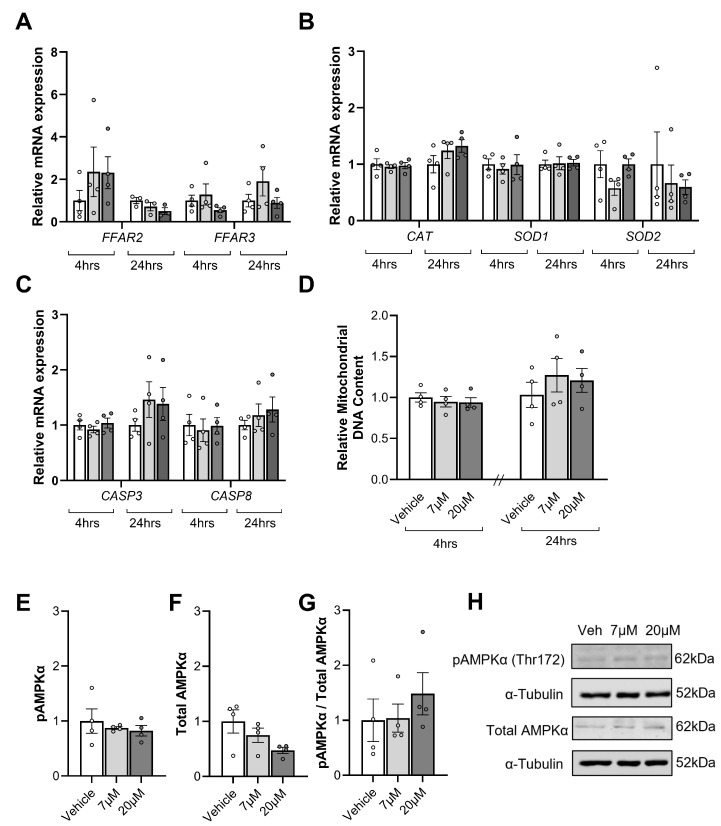
The influence of propionate treatments (7 µM and 20 µM) for 4 and 24 h on mRNA expression of the short-chain fatty acid (SFCA) receptors, free fatty acid receptor 2 (FFAR2), FFAR3 (**A**), the antioxidant genes catalase (CAT), superoxide dismutase 1 (SOD1), and SOD2 (**B**), and key regulators of apoptosis, caspase 3 (CASP3), and CASP8 (**C**) in placental explant tissue. Mitochondrial content at 4 and 24 h, as assessed by the ratio of markers of mitochondrial DNA divided by markers of nuclear DNA (**D**). Phosphorylated AMPKα (**E**), Total AMPKα (**F**) and Phosphorylated AMPKα per Total AMPKα (**G**) after 4 h were also assessed with representative blots shown above (**H**). Data are presented as mean ± SEM with a comparison between the vehicle (white), 7 µM propionate (light grey), and 20 µM propionate (dark grey). Individual dots represent the mean of technical replicates for four separate placentas that each received the three treatments. Statistical analysis included a repeated measures one-way ANOVA for parametric data or the Friedman test for non-parametric data. Significance was accepted when *p* < 0.05; blank spaces signify non-significance.

**Table 1 nutrients-17-01234-t001:** Maternal information and pregnancy outcomes for participants in adequate- (*n* = 25) and low- (*n* = 27) fibre diet groups.

		Adequate Fibre	Low Fibre	*p*
Clinical characteristics	SPRING Treatment (Placebo%) ^#^	56%	52%	ns
	Ethnicity (Caucasian%) ^#^	84%	85%	ns
	Baseline BMI (kg/m^2^) ^^^	30.86 (28.31–37.15)	30.88 (29.36–25.67)	ns
	28-week BMI (kg/m^2^) ^^^	33.76 (30.33–38.57)	33.75 (31.79–36.73)	ns
	Age (years)	33 (31–36)	30 (27–34)	*
	Fasting Blood Glucose (mmol/L)	4.20 (3.95–4.50)	4.20 ± (3.90–4.50)	ns
	C-Peptide (nmol/L) ^^^	0.80 (0.65–0.90)	0.80 (0.60–1.20)	ns
	Insulin (mU/L) ^^^	9.30 (6.45–12.00)	8.60 (6.00–16.00)	ns
	Cholesterol (mmol/L)	6.40 (5.45–7.25)	6.50 (6.20–7.70)	ns
	Triglycerides (mmol/L) ^^^	1.90 (1.50–2.70)	2.20 (1.60–2.60)	ns
	HDL (mmol/L)	1.90 (1.70–2.30)	1.80 (1.50–2.10)	ns
	LDL (mmol/L)	3.60 (3.03–4.25)	3.90 (3.30–5.00)	ns
	VLDL (mmol/L)	0.90 (0.70–1.10)	1.00 (0.70–1.20)	ns
Pregnancy	Gestational Age (weeks)	40.36 (39.00–41.22)	40.22 (38.57–41.18)	ns
Outcomes	Offspring Sex (Male%) ^#^	48%	44%	ns
	Birthweight (g)	3606 (3413–3977)	3833 (3437–4155)	ns
	Delivery (Vaginal%) ^#^	80%	66%	ns

^^^ denotes data analysed using the two-tailed Mann–Whitney U test and ^#^ Fisher’s exact test. Unmarked variables were analysed by the two-tailed unpaired *t*-test. * for *p* < 0.05 and ns for *p* > 0.05. Data are presented as median and interquartile range.

**Table 2 nutrients-17-01234-t002:** Maternal dietary information in adequate- (*n* = 25) and low- (*n* = 27) fibre diet groups.

Dietary	Energy Intake (kJ/day)^^^	5872 (4998–7350)	6102 (4586–6749)	*ns*
Data	Energy-adjusted fibre intake (mg/kJ)	3.83 (3.30–4.12)	2.09 (1.80–2.27)	****
	Gross fibre intake (g/day) ^^^	21.84 (16.86–30.05)	11.82 (9.58–15.46)	****
	Energy-adjusted carbohydrate intake (mg/kJ)	25.80 (24.57–27.83)	23.71 (20.99–24.84)	***
	Gross carbohydrate intake (g/day) ^^^	164.3 (132.2–205.3)	134.9 (108.7–167.6)	*
	Energy-adjusted fat intake (mg/kJ)	13.20 (11.75–15.65)	11.50 (10.40–13.20)	***
	Gross fat intake (g/day) ^^^	62.82 (51.56–74.18)	66.88 (51.97–73.75)	ns
	Energy-adjusted protein intake (mg/kJ)	11.33 (10.12–12.45)	11.93 (10.88–13.16)	ns
	Gross protein intake (g/day) ^^^	67.48 (54.48–83.52)	72.80 (48.82–94.40)	ns
	Energy-adjusted starch intake (mg/kJ)	13.15 (11.77–15.69)	11.48 (10.40–13.24)	***
	Gross starch intake (g/day) ^^^	80.01 (68.59–107.9)	68.08 (51.32–80.18)	*
	Energy-adjusted sugar intake (mg/kJ) ^^^	11.77 (10.24–13.51)	11.03 (8.22–13.13)	ns
	Gross sugar intake (g/day)	81.21 (58.16–96.81)	62.15 (44.54–89.16)	ns
	Energy-adjusted saturated fat intake (mg/kJ)	4.15 (3.61–4.75)	5.23 (4.77–5.59)	****
	Gross saturated fat intake (g/day)^^^	26.34 (20.52–31.51)	29.87 (24.72–33.87)	ns
	Energy-adjusted monounsaturated fat intake (mg/kJ) ^^^	3.51 (3.35–3.89)	3.79 (3.60–4.09)	*
	Gross monosaturated fat intake (g/day) ^^^	22.23 (18.04–26.56)	23.23 (16.94–26.68)	ns
	Energy-adjusted polyunsaturated fat intake (mg/kJ) ^^^	1.30 (1.15–1.75)	1.20 (1.00–1.300)	*
	Gross polyunsaturated fat intake (g/day) ^^^	9.06 (6.46–12.09)	6.59 (5.07–7.67)	**
	Energy from fat (%)	38.58 (36.34–40.84)	42.75 (39.51–44.90)	**
	Energy from carbohydrates (%)	43.82 (41.03–46.46)	39.59 (35.05–41.48)	****
	Energy from protein (%)	18.92 (16.89–20.79)	20.09 ± 0.64%	ns

^^^ denotes data analysed by THE two-tailed Mann–Whitney U test. Unmarked variables were analysed by the two-tailed unpaired *t*-test. * for *p* < 0.05, ** for *p* < 0.01 *** for *p* < 0.001, **** for *p* < 0.0001, and ns for *p* > 0.05. Data are presented as median and interquartile range.

**Table 3 nutrients-17-01234-t003:** Correlations between maternal diet and serum short-chain fatty acid levels at 28 weeks gestation. Spearman correlation coefficient (r). Significance accepted when *p* < 0.05 (shown in bold).

Macronutrient Intake (mg/kJ or g/Day)	Acetate		Propionate		Butyrate	
	r	*p*	r	*p*	r	*p*
Fibre (mg/kJ)	0.15	0.13	–0.08	0.76	0.17	0.28
Fibre (g/day)	0.09	0.55	–0.04	0.80	0.24	0.13
Carbohydrates (mg/kJ)	0.01	0.94	0.14	0.37	0.35	**0.03**
Starch (mg/kJ)	0.04	0.81	0.09	0.57	0.26	0.10
Sugars (mg/kJ)	0.02	0.86	0.08	0.61	0.18	0.27
Fat (mg/kJ)	0.04	0.61	0.09	0.57	0.26	0.10
Saturated fat (mg/kJ)	–0.08	0.61	–0.15	0.33	–0.38	**0.01**
Monounsaturated fat (mg/kJ)	–0.24	0.14	–0.26	0.10	–0.32	**0.04**
Polyunsaturated fat (mg/kJ)	0.01	0.93	0.13	0.41	0.26	0.09
Protein (mg/kJ)	0.10	0.51	–0.10	0.53	–0.15	0.34

## Data Availability

The data that support the findings of this study are not openly available due to reasons of sensitivity and are available from the corresponding author upon reasonable request. Data are located in controlled access data storage at The University of Queensland.

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
