# Peer review of "Dietary Fibre Modulates Gut Microbiota in Late Pregnancy Without Altering SCFA Levels, and Propionate Treatement Has No Effect on Placental Explant Function"

_nutrients, 2025, doi:10.3390/nu17071234_

Round 1

Reviewer 1 Report

Comments and Suggestions for Authors

In the paper “Dietary Fibre Modulates Gut Microbiota in Late Pregnancy Without Altering SCFA Levels, and Propionate Has No Effect on Placental Explant Function”, the authors compared faecal microbial composition and circulating SCFA concentrations at 28 weeks of gestation in overweight and obese pregnant participants who consumed either a low or adequate fibre diet. The research content is quite interesting. Here are some specific issues.

Comments:

Q1. The abstract lacks clarity in articulating the research objectives and presents incomplete results. It is recommended to enhance these aspects for greater precision and comprehensiveness. Besides, only overweight and obese pregnant women were included in the study, why?

Q2. Might other lifestyle and environmental factors besides dietary fiber intake also influence gut microbiota and SCFA levels?

Q3. Although 16S rRNA gene amplicon sequencing of the gut microbiota was performed, the presentation of the results appears to be less than comprehensive. For example, the relative content information of top10 or top20 strains was not reflected. Furthermore, why does Figure 1D not seem to show a group difference?

Q4. Only serum SCFA concentrations were measured; intestinal SCFA concentrations were not measured. There may be differences between the concentration of SCFA in the gut and serum, and the effect of SCFA in the gut on gut microbiota and host health may be more direct. It is recommended that the authors address the limitations of serum SCFA concentrations representing overall SCFA levels in the discussion section and consider adding measurements of intestinal SCFA concentrations in future studies to more fully understand the effects of dietary fiber on SCFA production.

Q5. Why was propionic acid chosen only for study in the placental explant experiment?

Q6. In Figure 4, why is the band longer in the veh group than in the other groups? Does this affect the analysis of the results? Also, what does the little circle in the bar graph represent, does it represent each parallel experiment?

Author Response

We thank the reviewers for their valuable time reviewing this manuscript. We have made a number of changes to the paper based on this feedback and feel that the paper is now much improved. In particular, we have expanded the limitations section significantly and included additional discussion based on the feedback from  the reviewers. When we could do so, we limited the number of additional words added to the manuscript to ensure that it remained as concise as possible but tried to make sure we addressed the key points as best as we could. Please see below the response to individual comments.

Reviewer 1

Q1. The abstract lacks clarity in articulating the research objectives and presents incomplete results. It is recommended to enhance these aspects for greater precision and comprehensiveness.

We have improved the abstract to increase clarity of the research aims and results.

Q2. Besides, only overweight and obese pregnant women were included in the study, why?

This study used samples collected as part of the SPRING cohort which only recruited participants that were overweight and obese people as the study aimed to look at the cohort where diet most likely influenced microbiome status. We have now commented on this further throughout.

Q3. Might other lifestyle and environmental factors besides dietary fiber intake also influence gut microbiota and SCFA levels?

The reviewer asks a very important question and we highlight that participants who consume a high fibre diet in this cohort, do so largely due to an increase in foods that are both high in total carbohydrates and fibre. They consume these foods at the expense of foods that are high in fat so carbohydrate and fat content are different between groups once you divide the cohort up by fibre intake. We have commented on this and now added a large section in the discussion related to these other factors.

Q4. Although 16S rRNA gene amplicon sequencing of the gut microbiota was performed, the presentation of the results appears to be less than comprehensive. For example, the relative content information of top10 or top20 strains was not reflected. Furthermore, why does Figure 1D not seem to show a group difference?

We have performed the microbiome analysis using a standard approach of firstly assessing how a factor has influenced diversity before focusing on specific genera. We have already included a correlation analysis of abundance vs various dietary factors including fibre for the top 50 most abundant bacterial genera. As fibre did not significantly impact many bacterial genera once the cohort was split into high and low fibre, we have only reported those that were closest to demonstrating a difference (top 7). Of these, only two were significant and the rest were not impacted by fibre intake. Figure 1D is one of many comparisons that we performed that show that fibre had minimal impact on bacterial diversity. We later discuss that we think this is due to the fact that the relative impact of fibre on bacterial populations is diminished in an environment so strongly influenced by pregnancy hormones and this supports many other studies by our group and others.

Q5. Only serum SCFA concentrations were measured; intestinal SCFA concentrations were not measured. There may be differences between the concentration of SCFA in the gut and serum, and the effect of SCFA in the gut on gut microbiota and host health may be more direct. It is recommended that the authors address the limitations of serum SCFA concentrations representing overall SCFA levels in the discussion section and consider adding measurements of intestinal SCFA concentrations in future studies to more fully understand the effects of dietary fiber on SCFA production.

We would anticipate that SCFA concentrations in fecal material would have a stronger relationship with the abundance of SCFA producing bacteria compared to levels found in blood. We have added a sentence into the discussion about how doing this would allow us to know how fibre intake in pregnancy impacts fecal SCFA content. The reason we didn’t include this analysis in the current study was that we were more interested in how fibre related SCFA content could impact tissues far away from the intestinal lumen. In particular, we needed to know how fibre could impact pregnancy related physiology that is tightly linked to placental function which is why we elected to study the level in the blood.

Q6. Why was propionic acid chosen only for study in the placental explant experiment?

Propionic acid was chosen as the SCFA to treat the placental explants due to it being the SCFA associated with the highest number of bacteria as well as it being previously linked to pregnancy complications. This is described in lines 334-337.

Q7. In Figure 4, why is the band longer in the veh group than in the other groups? Does this affect the analysis of the results? Also, what does the little circle in the bar graph represent, does it represent each parallel experiment?

We have performed these western blots on all of the experimental replicates (n=4 per group) and the data is consistent. We have shown a representative image of the blots in figure 4 and the entire blots are provided in supplementary files. The individual dots on the graphs represent the mean of technical replicates for four separate placentas.

Reviewer 2 Report

Comments and Suggestions for Authors

Dietary Fiber Modulates Gut Microbiota in Late Pregnancy Without Altering SCFA Levels, and Propionate Has No Effect on Placental Explant Function. 

In this study, the researchers compared fecal microbial composition and circulating SCFA concentrations at 28 weeks of gestation in overweight and obese pregnant participants who consumed either a low or adequate fiber diet. Also, placental explants were exposed to the SCFA, propionate. The researchers showed that maternal dietary fiber intake did not impact microbial diversity or richness but did impact the abundance of specific bacterial genera. 

SCFA concentrations did not differ between groups, but serum concentrations of acetate, propionate, and butyrate did correlate with the abundance of key bacterial genera. Propionate treatment of placental explants did not impact key pathways known to be impacted by SCFA in other tissues. 

This study demonstrates that dietary fiber has a modest impact on SCFA concentrations in pregnant women and that propionate does not impact key pathways in placental tissue. This suggests that previous associations between this SCFA and placental dysfunction may be due to other maternal factors.  

The abstract lacks experimental design, statistical analysis, and P value.

An excellent introduction.

The materials and methods section is well prepared. All methods are described in detail.

 L218:  adequate adequate-fiber diet group, remove adequate

L217-225: repeated results, revise.

Figures: this represents the mean +SEM or SD

Fig2: increase font size

Excellent discussion and conclusion

Author Response

We thank the reviewers for their valuable time reviewing this manuscript. We have made a number of changes to the paper based on this feedback and feel that the paper is now much improved. In particular, we have expanded the limitations section significantly and included additional discussion based on the feedback from  the reviewers. When we could do so, we limited the number of additional words added to the manuscript to ensure that it remained as concise as possible but tried to make sure we addressed the key points as best as we could. Please see below the response to individual comments.

Reviewer 2

  1. The abstract lacks experimental design, statistical analysis, and P value.

See comments for reviewer 1. We have improved the abstract while keeping in line with the requirements for the journal abstract style. Due to tight world limits, we have not added in statistical analysis or p values in the abstract.

  1. An excellent introduction.

We thank the reviewer for this comment.

  1. The materials and methods section is well prepared. All methods are described in detail.

We thank the reviewer for this comment.

  1. L218:  adequate adequate-fiber diet group, remove adequate

We have now fixed this mistake.

  1. L217-225: repeated results, revise.

We have checked through the results and corrected any other errors.

  1. Figures: this represents the mean +SEM or SD

This is SEM and we have updated the figure legend to reflect this

  1. Fig2: increase font size

We have rotated the image to ensure the font size is large enough.

  1. Excellent discussion and conclusion

We thank the reviewer for this comment.

Reviewer 3 Report

Comments and Suggestions for Authors

Areas for Improvement and Limitations:

  1. Dietary Assessment:
    • DQESv2 Limitations: The Dietary Questionnaire for Epidemiological Studies (DQESv2) relies on self-reporting, which can introduce bias. Consider discussing the limitations of this method in the discussion.
    • Confounding Dietary Factors: Table 1 highlights significant differences in macronutrient intake between the groups, not just fiber. This makes it difficult to isolate the specific effect of fiber. The discussion must be expanded to discuss the confounding effects of these other dietary differences.
    • Dietary Quality: While fiber is examined, overall dietary quality could be assessed. A score that measures the quality of the diet may add value to the research.
  2. Microbiome Analysis:
    • 16S rRNA Limitations: 16S rRNA sequencing provides information on bacterial composition but not function. Consider acknowledging this limitation. Metagenomics could provide a more comprehensive view.
    • Calypso Web-Server: The Calypso Web-Server is no longer available. This should be addressed, and if possible, the analysis should be redone using a current available tool.
    • qPCR Discrepancies: The lack of correlation between qPCR and 16S rRNA data for Faecalibacteriumneeds further investigation and discussion. Possible reasons should be explored, such as primer specificity.
  3. SCFA Analysis:
    • Circulating vs. Fecal SCFA: Measuring circulating SCFA may not fully reflect gut SCFA production. Fecal SCFA measurements could provide a more direct assessment of microbial fermentation.
    • SCFA Variability: SCFA levels can fluctuate significantly. Consider discussing factors that may influence SCFA variability, such as timing of sample collection.
  4. Placental Explant Experiments:
    • Physiological Relevance: The chosen propionate concentrations (7µM and 20µM) are discussed, but further justification for their physiological relevance in the placental tissue is needed. The discussion needs to include more information about the previous research that used higher dosages.
    • Limited SCFA: Only propionate was tested. Testing other SCFAs (acetate, butyrate) would provide a broader understanding of SCFA effects on the placenta.
    • Sample Size: The placental explant experiments used tissue from only four placentas, which is a small sample size. This limits the generalizability of the findings.
    • Time Points: Only 4 and 24 hour time points were used. Other time points could show different results.
    • Placental Heterogeneity: Placental tissue is heterogeneous. The methods section does state that the tissue was homogenized, but this should be discussed as a possible limitation.
  5. Statistical Analysis:
    • Multiple Comparisons: With multiple correlations and comparisons, consider addressing the issue of multiple testing and potential type I errors.
    • Confounding variables: The age difference between the two groups, should be considered as a confounding variable. If possible, a statistical test should be performed to remove the age difference as a contributing factor.
  6. Writing and Presentation:
    • Abstract: The abstract could be more concise and highlight the key findings more effectively.
    • Introduction: The introduction is good, but could be more focused.
    • Discussion: The discussion should more thoroughly address the limitations of the study and the clinical implications of the findings.
    • Figure Clarity: Ensure all figures and tables are clearly labeled and easy to understand.
    • Table 1: The table is very large. Consider breaking it down into smaller, more focused tables.
  7. Ethical Considerations:
    • While the study was ethically approved, consider adding a statement about data sharing and accessibility in compliance with open science principles.

Specific Recommendations:

  • Strengthen the discussion of the confounding effects of macronutrient intake.
  • Provide more in-depth discussion of the limitations of 16S rRNA sequencing and the DQESv2.
  • Justify the choice of propionate concentrations for the placental explant experiments.
  • Increase the sample size for placental explant experiments if possible.
  • Address the discrepancy between qPCR and 16S rRNA data for Faecalibacterium.
  • Re-run the data that was ran through the calypso web server, using a current tool.
  • Consider adding a statement about data sharing and accessibility.
  • Add a statement about the limitations of the study

Author Response

Reviewer 3

We thank the reviewers for their valuable time reviewing this manuscript. We have made a number of changes to the paper based on this feedback and feel that the paper is now much improved. In particular, we have expanded the limitations section significantly and included additional discussion based on the feedback from  the reviewers. When we could do so, we limited the number of additional words added to the manuscript to ensure that it remained as concise as possible but tried to make sure we addressed the key points as best as we could. Please see below the response to individual comments.

  1. DQESv2 Limitations: The Dietary Questionnaire for Epidemiological Studies (DQESv2) relies on self-reporting, which can introduce bias. Consider discussing the limitations of this method in the discussion.

We have added in a detailed discussion about the limitations of self-reporting of dietary information and how this may impact study conclusions. 

  1. Confounding Dietary Factors: Table 1 highlights significant differences in macronutrient intake between the groups, not just fiber. This makes it difficult to isolate the specific effect of fiber. The discussion must be expanded to discuss the confounding effects of these other dietary differences.

We set out to assess how fibre impacted outcomes in this study and yet by separating participants by fibre intake, there are significant changes in carbohydrates and fats that likely also contribute to the findings of this study. We had previously considered renaming these groups as a low quality and a high quality diet based on these changes but we don’t believe that either group can be classified as high or low quality, particularly given that we intentionally selected patients based on fibre.  We have added much to the discussion about this point.  

  1. Dietary Quality: While fiber is examined, overall dietary quality could be assessed. A score that measures the quality of the diet may add value to the research.

As per the response above, we did consider if our analysis was indicative of dietary quality but given that we separated groups based on fibre, not an overall diet quality assessment, we have kept the analysis based on fibre and discussed the fact that it is very hard to determine the effects of fibre in isolation without changes to other dietary factors. We have previously looked at overall dietary quality in a different subset of women enrolled in the SPRING study and how it impacts bacterial diversity but that study was not focused on fibre and so addresses a different question (O’Connor et al. J Human Nutrition and Dietetics, 2023; 36: 1425. https://doi.org/10.1111/jhn.13123). However, that study showed that dietary quality is not associated with overall changes to the gut microbiota. We again have discussed the limitations of trying to separate our groups based on fibre when other dietary factors remained different between groups.

  1. 16S rRNA Limitations: 16S rRNA sequencing provides information on bacterial composition but not function. Consider acknowledging this limitation. Metagenomics could provide a more comprehensive view.

We acknowledge that 16S rRNA sequencing focuses on composition but not function. We have now highlighted that additional information related to function would be a highly informative future direction

  1. Calypso Web-Server: The Calypso Web-Server is no longer available. This should be addressed, and if possible, the analysis should be redone using a current available tool.

Unfortunately the Calypso Web-Server is no longer available. We had indicated this in the submitted manuscript. We now further highlight in the discussion that this is a limitation but repeating the analysis using a new tool is beyond the scope of the current study.

  1. qPCR Discrepancies: The lack of correlation between qPCR and 16S rRNA data for Faecalibacterium needs further investigation and discussion. Possible reasons should be explored, such as primer specificity.

The QPCR did successfully validate the 16s sequencing data for three of the four bacterial genera assessed. The reviewer rightly highlights that the qPCR amplification of the sequence detected by the Faecalibacterium primers did not correlate with the 16s rRNA data. This is likely due to the fact that there are many species of Faecalibacterium and our primers are very specific for just one species of Faecalibacterium (F. Duncaniae). F. Duncaniae is also not the most common Faecalibacterium species (F. prauznitzii).  We have discussion related to this in the limitations section of the discussion.

  1. Circulating vs. Fecal SCFA: Measuring circulating SCFA may not fully reflect gut SCFA production. Fecal SCFA measurements could provide a more direct assessment of microbial fermentation.

See response to reviewer 1: Fecal SCFA meaurements would have added a new angle to the manuscript and allowed better understanding of the impact of diet of production of SCFA rather than levels detectable in blood. However this was not part of the aims of the study, which focused on the effects of circulating SCFA in the host,  and we have included this as a limitation.

  1. SCFA Variability: SCFA levels can fluctuate significantly. Consider discussing factors that may influence SCFA variability, such as timing of sample collection.

We thank the reviewer for this point and we have added discussion around this to the manuscript. It would have been highly informative to collect samples for analysis of SCFA at additional  timepoints from these women.

  1. Physiological Relevance: The chosen propionate concentrations (7µM and 20µM) are discussed, but further justification for their physiological relevance in the placental tissue is needed. The discussion needs to include more information about the previous research that used higher dosages.

We have now added significant discussion around this to the paper. The doses selected are based on levels commonly detected in circulation and are likely the doses that the placenta will be exposed when in the body. We have highlighted that a limitation of the current study is that we did not include a “positive control”, which may have been treatment at the extremely high dose that have been reported in the literature previously which better reflect SCFA levels seen in the intestinal contents. We have included the findings from studies which have treated placental cells and tissues with higher doses of SCFA.

  1. Limited SCFA: Only propionate was tested. Testing other SCFAs (acetate, butyrate) would provide a broader understanding of SCFA effects on the placenta.

We only had capacity to assess the impact of one SCFA on placental explants. We selected propionate based on our findings from the current study and work previously conducted by others. It would have been ideal to repeat these experiments with acetate and butyrate to get a more complete understanding of how SCFA at physiological doses impact placental outcomes. We have included this in the limitations and future directions.

  1. Sample Size: The placental explant experiments used tissue from only four placentas, which is a small sample size. This limits the generalizability of the findings.

We do acknowledge that this is a small sample size and have included this in the new limitation section.

  1. Time Points: Only 4 and 24 hour time points were used. Other time points could show different results.

We do acknowledge that additional time points would have allowed for a more in depth analysis and we have included this in the new limitation section. However, 4 and 24 hours provide sufficient time to observe effects on mRNA and protein levels which why these were selected in this case. We have added some discussion to duration of exposure to the paper.

  1. Placental Heterogeneity: Placental tissue is heterogeneous. The methods section does state that the tissue was homogenized, but this should be discussed as a possible limitation.

Placental tissue is heterogeneous and subtle differences in cellular composition of the explanted tissue may have contributed to some variability in the data produced. We did collect placental tissue from 5 sites of the placenta and  dissected these into 2mm3 segments. We then pooled 120mg worth of these small segments together making sure that we had tissue from each dissected region mixed together.  . This is a standard approach used to limit selecting a biased sample site.

  1. Multiple Comparisons: With multiple correlations and comparisons, consider addressing the issue of multiple testing and potential type I errors.

We acknowledge this as a limitation.

  1. Confounding variables: The age difference between the two groups, should be considered as a confounding variable. If possible, a statistical test should be performed to remove the age difference as a contributing factor.

We acknowledge this as a limitation. However, while the difference between the groups is statistically significant, its clinical significance should not be overstated e.g. difference of 3 years.

  1. Abstract: The abstract could be more concise and highlight the key findings more effectively.

We have reworded the abstract and carefully tried to balance this requested change with that of reviewer one and two.

  1. Introduction: The introduction is good, but could be more focused.

We have reworded the introduction slightly to make it more focused.

  1. Discussion: The discussion should more thoroughly address the limitations of the study and the clinical implications of the findings.

This is where we have made the most significant changes to the paper based on the feedback of reviewers. We have discussed all of the limitations of the current paper and highlighted future directions.

  1. Figure Clarity: Ensure all figures and tables are clearly labelled and easy to understand.

We have improved the readability of figure 2.

  1. Table 1: The table is very large. Consider breaking it down into smaller, more focused tables.

We have split table one into two tables. This may also help prevent confusion between what we had called maternal characteristics and maternal food intake

  1. While the study was ethically approved, consider adding a statement about data sharing and accessibility in compliance with open science principles.

This statement was originally included in the section of the submission process where this is normally added. Data will be available upon reasonable request to the authors, this is to comply with the ethical permissions obtained from our participants.